# Identification and Characterization of Jasmonic Acid Biosynthetic Genes in *Salvia miltiorrhiza* Bunge

**DOI:** 10.3390/ijms23169384

**Published:** 2022-08-20

**Authors:** Xiaoshan Xue, Runqing Li, Caijuan Zhang, Wenna Li, Lin Li, Suying Hu, Junfeng Niu, Xiaoyan Cao, Donghao Wang, Zhezhi Wang

**Affiliations:** 1Key Laboratory of the Ministry of Education for Medicinal Resources and Natural Pharmaceutical Chemistry, National Engineering Laboratory for Resource Development of Endangered Crude Drugs in Northwest of China, Shaanxi Normal University, Xi’an 710062, China; 2Institute of Landscape Science, Taiyuan University, Taiyuan 030032, China

**Keywords:** jasmonic acid, *Salvia miltiorrhiza*, secondary metabolism, traditional Chinese medicine

## Abstract

Jasmonic acid (JA) is a vital plant hormone that performs a variety of critical functions for plants. *Salvia miltiorrhiza* Bunge (*S. miltiorrhiza*), also known as Danshen, is a renowned traditional Chinese medicinal herb. However, no thorough and systematic analysis of JA biosynthesis genes in *S. miltiorrhiza* exists. Through genome-wide prediction and molecular cloning, 23 candidate genes related to JA biosynthesis were identified in *S. miltiorrhiza*. These genes belong to four families that encode lipoxygenase (LOX), allene oxide synthase (AOS), allene oxide cyclase (AOC), and 12-OPDA reductase3 (OPR3). It was discovered that the candidate genes for JA synthesis of *S. miltiorrhiza* were distinct and conserved, in contrast to related genes in other plants, by evaluating their genetic structures, protein characteristics, and phylogenetic trees. These genes displayed tissue-specific expression patterns concerning to methyl jasmonate (MeJA) and wound tests. Overall, the results of this study provide valuable information for elucidating the JA biosynthesis pathway in *S. miltiorrhiza* by comprehensive and methodical examination.

## 1. Introduction

Jasmonic acid (JA) and its derivatives (e.g., jasmonic isoleucine (JA-Ile) and methyl jasmonate (MeJA)) are fatty acids derived from cyclopentanone and belong to the oxidized lipid family [1]. In addition to promoting leaf senescence, JA can inhibit root growth, accelerate fruit ripening, and regulate the development of flowering plants [2,3]. It is also a signal transduction molecule that has been linked to plant resistance to biological stress (injuries, insect bites, etc.), abiotic stress (drought, low temperatures), and plant hormone induction, which have been reported in detail [1,4,5].

JA is synthesized from α-linolenic acid (α-LeA), which is catalyzed in the chloroplast and transported to peroxisomes via a series of oxidation and reduction reactions (Figure 1) [6]. Firstly, 13-hydroperoxylinolenic acid (13-HPOT) is synthesized from α-LeA via lipoxygenase (LOX), which is non-heme iron-containing oxygenase that catalyzes the oxygenation of polyunsaturated fatty acids to produce hydroperoxides that play important roles in plant growth and development, as well as in response to abiotic and biological stresses [7]. Secondly, allene oxide synthase (AOS) generates unstable allylic epoxides that hydrolyze spontaneously in water to 13-hydroxy-12-oxo-octadecadienoic acid (α-ketols) and 9-hydroxy-12-oxo-octadecadienoic acid (γ-ketols), and cyclize to a racemic mixture of *cis*-(+) and *cis*-(−) enantiomers [8,9]. It catalyzes the production of 12S, 13S-epoxy-linolenic acid (12,13-EOT) using 13-HPOT as the substrate in chloroplasts. Subsequently, allene oxide cyclase (AOC) can approach AOS products and rearrange unstable 12,13-EOT to create pentenyl and carboxylic acid side chains at positions 3 and 7 of the pentanone ring, respectively, resulting in racemized 12-oxo-phydienoic acid (12-OPDA) [10,11]. Only *cis*-(+)-12-OPDA can be further converted to natural JA. Through the subsequent steps, the 12-OPDA that is transported to the peroxisome and cyclopentenone ring is catalyzed by 12-OPDA reductase3 (OPR3) [12], after which JA is generated via a reduction reaction that is catalyzed by three β-oxidation cycles [13]. Finally, JA diffuses into the cytoplasm and forms various derivatives through different modifications [14].

*Salvia miltiorrhiza* Bunge (*S. miltiorrhiza*) is a renowned medicinal plant that belongs to labiaceae [15], which has been used to treat various diseases, particularly coronary artery and cerebrovascular conditions [16]. For many years, people have used the root or rhizome of *S. miltiorrhiza* as a medication [17], with its main medicinal ingredients primarily including two compounds, water-soluble salvianolic acid and fat-soluble tanshinone [18]. Studies have indicated that putting inducers such as MeJA, GA_3_, IAA, and Ag^+^ can effectively alter the amount of tanshinone or phenolic acids in *S. miltiorrhiza* [19,20], whereas MeJA has the ability to simultaneously enhance the accumulation of both chemicals [21]. Additionally, MYB, bHLH, WRKY, COI1, and other transcription factors and proteins that respond to the JA signal can encourage *S. miltiorrhiza* secondary metabolism [22,23,24,25]. The JA signalling pathway is the most significant to affect the secondary metabolic accumulation of *S. miltiorrhiza*. Therefore, this study systematically analyzed the four gene families that facilitated JA biosynthesis in *S. miltiorrhiza*. This laid a foundation for exploring the kinetics involved in its biogenesis in medicinal plants, as well as the internal mechanisms of resistance against external stress.

## 2. Results and Discussion

### 2.1. Prediction and Molecular Cloning of JA Biosynthesis-Related Genes in S. miltiorrhiza

Using the BioEdit BLASTP program and the LOX, AOS, AOC, and OPR3 proteins as queries, 23 presumed homologous genes were surveyed from the genome database of *S. miltiorrhiza*, including 9 *SmLOXs*, 7 *SmAOSs*, 2 *SmAOCs*, and 5 *SmOPR3s*. After excluding sequences without conserved domains, 23 genes were designated as *SmLOX1*-*SmLOX9*, *SmAOS1*-*SmAOS7*, *SmAOC1*-*SmAOC2*, and *SmOPR3-1**-SmOPR3-5* according to different families. As shown in Appendix A, the four gene families’ sequence length were statistically listed, and the fundamental traits of the four gene families were predicted, including protein molecular weight (MW)/length, isoelectric point (pI), and subcellular localization. All these nucleotide sequences were uploaded to NCBI and obtained GenBank accession numbers. Full-length design primers for each sequence were selected and listed in Appendix A for PCR identification. The results of molecular cloning were sequenced and identified; thus, all the genes were analyzed in the next step.

### 2.2. SmLOX

Numerous plants have been used to study LOXs through cloning and enzyme identification, such as *Arabidopsis* [26], rice (*Oryza sativa*) [27], cotton (*Gossypium hirsutum*) [28], tomato [29], soybean (*Glycine max*) [30], maize (*Zea mays*) [31], and pepper [32]. According to the different sites of carbon atoms where their substrates might combine with oxygen, LOXs can be separated into 9-LOXs and 13-LOXs [33]. Six LOXs were found in *Arabidopsis*, four of which were 13-LOXs (AtLOX2, AtLOX3, AtLOX4, AtLOX6), while the other two were 9-LOXs (AtLOX1, AtLOX5), which were involved in cell formation and death [34]. Only 13-LOXs were involved in JA synthesis, which exhibited specific activities in injuries, wound responses, induction of senescence, fertility, and flower development [3]. In particular, AtLOX6 is crucial for the formation of JA in roots during *Arabidopsis**’* resistance to biotic and abiotic stresses [35]. While AtLOX3 is capable of withstanding salt stress [36], AtLOX4 takes involvement in defenses against root knot nematodes [37]. The majority of research on LOXs have observed that a portion of this gene family can be activated by JAs to boost their accumulation and synthesis, or they can stimulate the synthesis of JA and other compounds at injury sites, including tomato [38], cotton [39], and banana [40], among others.

In this study, nine *SmLOX* genes were found, and the lengths of their proteins ranged from 906 bp to 499 bp. Subcellular localization results showed that most SmLOXs resided in the cytoplasm (Appendix A). Although SmLOX6, SmLOX7, and SmLOX9 only had a lipoxygenase domain, other SmLOXs were anticipated to also include a PLAT domain (Appendix A). SmLOX5 and SmLOX9 were shown to have strong homology and to be clustered together in a phylogenetic tree with the LOX family of genes from multiple species, whereas SmLOX2 and SmLOX3 had high homology and were closely related to AtLOX5 (Figure 2). Meanwhile, SmLOX1, SmLOX6, SmLOX7, and SmLOX8 were clustered with AtLOX1, AtLOX6, AtLOX2, AtLOX3, and AtLOX4 in *Arabidopsis*, respectively. One cluster included only SmLOX4 molecules. The results suggested that these SmLOXs may play different roles in *S. miltiorrhiza**,* just as the LOXs in *Arabidopsis*. *SmLOX2-SmLOX3* and *SmLOX5-SmLOX9* gene pairs underwent Ka/Ks analysis to determine whether the homologous genes have been screened and purified (Appendix A). The results revealed that the Ka/Ks ratio of the two gene pairs was less than 1, indicating that selective purification was putting pressure on them [41]. Further sequence analysis showed that the SmLOX family’s motifs were all conserved and comprised introns (Figure 3). In addition, the conserved lipoxygenase domain of SmLOXs revealed a high degree of overlap in multiple sequence alignment (Appendix A).

The *SmLOX* expression patterns were different between the roots, stems, leaves, and flowers. *SmLOX1*, *SmLOX2*, *SmLOX5*, and *SmLOX9* were all significantly expressed in flowers, roots, stems, and leaves, respectively, while *SmLOX7* was negligibly expressed in roots (Figure 4). Using the promoter analysis of the first 1000 bp, multiple *SmLOX*s were found to contain *cis*-elements that were sensitive to MeJA and wound stress (Figure 5) (Appendix A). Once plant leaves were damaged, the responses of multiple *SmLOX*s changed in a variety of ways throughout time (Figure 6). Among them, *SmLOX2* and *SmLOX8* showed a considerable rise in expression within 1 h; *SmLOX9* increased from 6 h to 24 h; *SmLOX1* initially increased and subsequently declined; while other *SmLOX*s exhibited an unstable variable pattern. The expressions of *SmLOX2*, *SmLOX4*, and *SmLOX5* initially decreased and then increased when the plants were exposed to MeJA (Figure 7), while *SmLOX6* and *SmLOX8* first increased and then decreased, with *SmLOX7* and *SmLOX9* having the highest expressions at 1 h and 3 h, respectively. The biological process of *SmLOX*s was predicted through oxylipin biosynthesis, fatty acid biosynthesis, and lipid oxidation, according to GO annotation data. Molecular level functions consisted of two processes; oxidoreductase activity, which acted on single donors via the incorporation of two oxygen atoms, and metal ion binding. The cellular component refers primarily to the cytoplasm (Figure 8) (Appendix A).

### 2.3. SmAOS

In both dicotyledonous and monocotyledonous plants, AOSs have been extensively studied [42], with the number of AOS families varying between species. For example, one AOS gene was reported in *Arabidopsis* [43], flax (*Linum usitatissimum* L.) [44], and barrel medic (*Medicago truncatula*) [45]. Two AOS genes have been reported in barley (*Hordeum vulgare*) [46] and tomato (*Solanum lycopersicum*) [47,48], with three in potato (*S. tuberosum*) [49]. AOSs are a class of CYP74 enzymes [50], which includes divinyl ether synthase (DES) [51], hydroperoxide lyase (HPL, synonym hemiacetal synthase) [52], and epoxyalcohol synthase (EAS) [53]. Interestingly, several CYP74 enzymes possess dual HPL/EAS or triple HPL/EAS/AOS activities [54,55].

Seven *SmAOS*s were identified, with protein lengths ranging from 533 bp to 271 bp, and predicted to reside within the chloroplast or cytoplasm, except for SmAOS7 (Appendix A). In contrast to SmAOS7, which had a CYP74 structure, SmAOS1-SmAOS3 proteins had conserved PLN02468, SmAOS4-SmAOS6 domains, which were predicted to contain a p450 super family structure (Appendix A). According to the phylogenetic tree results, SmAOS1, SmAOS2, and SmAOS6 were clustered in one branch; SmAOS4, SmAOS3, and SmAOS5 were clustered in one branch, and SmAOS7 was isolated in one branch (Figure 2). Indicating that these homologous genes were subject to intense purification selection pressures during their development, the Ka/Ks for *SmAOS2*-*SmAOS6* and *SmAOS3*-*SmAOS5* were both lower than 1 (Appendix A). Sequence analysis indicated that only *SmAOS3* and not the other *SmAOSs* contained an intron. SmAOS6 and SmAOS7 possessed two conserved motifs, SmAOS5 had three different conserved motifs, while all other SmAOSs contained five conserved motifs, as listed above (Figure 3). Multiple sequence alignment results further demonstrated the non-uniformity of the SmAOS family sequences (Appendix A).

The distinct *SmAOS*s tissue expression profiles of revealed that *SmAOS7* was highly expressed in leaves; *SmAOS1* was highly expressed in stems; *SmAOS3* and *SmAOS5* patterns were abundantly expressed in both roots and leaves; and other *SmAOSs* were highly expressed in roots (Figure 4). On the *SmAOS* promoter, multiple MeJA and other hormone response elements were found (Figure 5, Appendix A). Following the MeJA treatment, *SmAOS3* and *SmAOS5* were highly expressed at 12 h, and *SmAOS7* was highly expressed at 10 min. The expression trends of *SmAOS2*, *SmAOS4*, and *SmAOS6* were similar, and the expression of *SmAOS1* initially decreased and then increased (Figure 7). When the leaves were injured, *SmAOS1*-*SmAOS6* exhibited comparable corresponding trends, with peak values at 1 h, while *SmAOS7* had divergent reaction variations (Figure 6). It was projected that all *SmAOS*s would take part in the sterol metabolic process, with *SmAOS1*, *SmAOS2*, and *SmAOS6* participating in JA biosynthesis. All *SmAOS*s had oxidoreductase and monooxygenase activities, in addition to binding for iron ions and heme. Furthermore, *SmAOS1* and *SmAOS2* exhibited AOS activity functions, while *SmAOS3*, *SmAOS4*, *SmAOS5*, and *SmAOS7* had lyase activity functions (Figure 8, Appendix A).

### 2.4. SmAOC

AOC was localized to plastids such as AOS [11] and it was believed that the two enzymes reacted through close contact [6]. Small gene families and low copy levels of AOC gene families were present in *Arabidopsis* [10], soybean [56], cotton (*Gossypium hirsutum*) [57], double copies in *Medicago truncatula* [58] and watermelon [59], and single copies in tomato (*Solanum lycopersicum*) [11] and rice [60]. Four AtAOCs had particular tissue and organ expression in *Arabidopsis* [61]. The heteroplasmic control of AtAOCs trimer activities were found in vitro and in vivo, and its function was demonstrated via site-directed mutagenesis [62]. In general, the JA levels did not increase when *AOS* or *AOC* were overexpressed, as the required α-LeA substrate was produced only in response to external stimuli (e.g., injury) but could only be significantly stimulated after injuries [63].

When the sequences in the *S. miltiorrhiza* database were analyzed, only two *SmAOC*s were discovered, among which *SmAOC1* had already been described [64]. The two SmAOC proteins had lengths of 246 bp and 239 bp, and both were found on the chloroplast (Appendix A). They both had complete Allene_ox_cyc domains (Appendix A). However, both did not cluster with AtAOCs, and their evolutionary findings did not resemble one another. SmAOC1 was found to be clustered into a single branch, while SmAOC2 was close to IiAOC, NtAOC, SlAOC, and StAOC (Figure 2). Additionally, the conserved motifs and multiple sequence alignments contained in SmAOC1 and SmAOC2 were remarkably similar, despite the fact that *SmAOC1*’s genome sequence possessed an intron, while that of *SmAOC2*’s contained two introns (Figure 3 and Appendix A).

*SmAOC1* and *SmAOC2*’s expressions were analyzed, and it was discovered that *SmAOC1* was primarily highly expressed in leaves while *SmAOC2* expression was marginally greater in stems, with no discernible difference in expression in any other plant components (Figure 4). On the 1000 bp promoter, more light responsive elements were discovered; however, no MeJA responses or other hormone-related *cis*-elements were found on this region (Figure 5, Appendix A). Nevertheless, it was found that *SmAOC1* and *SmAOC2* responded similarly to MeJA treatment and stress injured in *S. miltiorrhiza* seedlings (Figure 6 and Figure 7). Interestingly, in terms of the speed of response, there was a temporal succession effect in their responses, which may have been due to some antagonistic regulation. Their GO annotation results were entirely consistent, showing that both were involved in JA biosynthesis and mediated signaling pathway. They responded to various light conditions, vitamin B2, insect stress, and induced the plant to develop resistance. They were also involved in the regulation of plant growth and development and the flowering cycle (Figure 8, Appendix A).

### 2.5. SmOPR3

There were more than three OPRs in *Arabidopsis* and tomato; however, only OPR3 was specific for *cis*-(+)-OPDA [12,65]. To date, all OPR3 clones have been shown to carry peroxisome target sequences, and the OPR proteins were localized to peroxisome sites [12]. Experimental evidence of the effects of an independent mutation of OPR3 in *Arabidopsis* has been shown [66,67], where the mutant plant opr3-1 was typically employed for distinguishing JA and OPDA specific responses [68]. Studies in rice demonstrated that only OsOPR7 was involved in JA synthesis, which could compensate for opr3 mutants in *Arabidopsis**’* functional loss [69]. Since the OPDA transporter of the chloroplast membrane is yet unclear, a parallel pathway to input OPDA via ion capture has been proposed [70].

The *S. miltiorrhiza* database contained five homologous *SmOPR3* genes whose proteins ranged in length from 426 bp to 239 bp. All SmOPR3s, with the exception of SmOPR3-5, were localized in the chloroplast or cytoplasm, according to subcellular localization results (Appendix A). Each and every SmOPR3s had all of Oxidored_FMN conserved domains (Appendix A). The phylogenetic tree showed that only SmOPR3-1 was closely related to AtOPR3, whilst the other four SmOPR3s and PsOPR3 were closely related to OsOPR3 and ZmOPR3 of monocotyledons (Figure 2). Given that *SmOPR3-2* and *SmOPR3-3* are homologous pairs and have Ka/Ks values below 1, it is apparent that selective purification also applies to them (Appendix A). All SmOPR3s possessed the same four motifs and sequence arrangements, meanwhile *SmOPR3-1* to *SmOPR3-4* had similar gene structures, except for *SmOPR3-5* (Figure 3).

*SmOPR3*s exhibit various expression patterns in different tissues. *SmOPR3-1* and *SmOPR3-4* are primarily highly expressed in roots; *SmOPR3-2* and *SmOPR3-3* are highly expressed in stems; and *SmOPR3-5* is equally expressed in all tissues (Figure 4). The promoter sequence of several *SmOPR3*s was predicted to contain MeJA response elements, and WUN-motif was also found in the promoter of *SmOPR3-5* (Figure 5, Appendix A). All *SmOPR3*s’ expression levels peaked at 3 h after MeJA administration and showed a trend of first increasing and then decreasing levels (Figure 7). Following injury stress, *SmOPR3-2* to *SmOPR3-5* showed the same response trend, where *SmOPR3-4* had the most notable response (Figure 6). The sole difference was that *SmOPR3-1* first exhibited an upward-trend before switching to a downward-trend. All five *SmOPR3*s genes were involved in oxylipin and fatty acid biosynthesis. *SmOPR3-1* to *SmOPR3-4* participated in the biosynthesis of JA as well. Furthermore, *SmOPR3-1* was associated with stamen development and fungus responses. In terms of molecular functionality, all *SmOPR3* had FMN binding capacities, where *SmOPR3-1* and *SmOPR3-5* exhibited 12-oxophytodienoate reductase activities, while the other three *SmOPR3*s had oxidoreductase activities. *SmOPR3-1*, in contrast to the other *SmOPR3*s, also possessed the same capacity for protein binding (Figure 8, Appendix A).

## 3. Materials and Methods

### 3.1. Plant Materials and Treatments

*Salvia miltiorrhiza* Bunge (line 993) plants with full genomic sequences [71] were grown in a field nursery at the Institute for Medicinal Plant Development (IMPLAD). Fresh roots, stems, leaves, and flower tissues were collected from two-year-old *S. miltiorrhiza* plants, when the aboveground components of *S. miltiorrhiza* grew vigorously. The preparation of the plants and media for the MeJA treatments has been described previously [72]. MeJA (100 μM) was dissolved in a 10% ethanol solution and sprayed evenly on the leaves of the plants, while a control test was conducted using a solvent. Injury stress was induced on the leaves by making multiple incision wounds of similar size and using normal untreated plants as a control. Following the different treatments, the leaf and root tissues of plants of similar sizes were collected at eight time points, as 0 h, 10 min, 30 min, 1 h, 3 h, 6 h, 12 h, and 24 h, respectively. All samples were immediately frozen in liquid nitrogen and stored at −80 °C for pending RNA isolation. Each group of samples had three independent biological replicates.

### 3.2. Gene Prediction and Molecular Cloning

The protein sequence of the JA biosynthesis gene in different species was obtained from the NCBI website (https://www.ncbi.nlm.nih.gov/, accessed on 22 September 2021), whereas the related homologous genes of the JA biosynthesis gene in the *S. miltiorrhiza* gene bank were compared and screened using BioEdit (Los Altos, CA, USA) software. Subsequently BLAST (https://blast.ncbi.nlm.nih.gov/Blast.cgi?PROGRAM=blastp&PAGE_TYPE=BlastSearch&LINK_LOC=blasthome, accessed on 22 September 2021) comparison was performed on all candidate sequences to ensure screening accuracy [73]. All genes were designed with cloned primers whose sequences are listed in Appendix A. Using cDNA as template, most genes were successfully cloned by the PCR amplification method and verified by sequencing.

### 3.3. Sequence Features and Prediction of Subcellular Locations

The lengths of cDNA, DNA, and protein sequences of all candidate genes were calculated and listed (Appendix A). The theoretical molecular weight (*M*_w_) and isoelectric point (pI) were predicted using the Compute pI/*M*_w_ tool on the ExPASy server (http://web.expasy.org/protparam/, accessed on 22 September 2021) [74]. Conservative domains were looked up in the Pfam database (http://pfam.xfam.org/family/PF00305, accessed on 22 September 2021) [75] and NCBI Batch CD-search [76]. Subcellular localization was predicted using the WOLF PSORT (https://www.genscript.com/wolf-psort.html?src=leftbar, accessed on 22 September 2021) tool [77].

### 3.4. Multiple Sequence Alignments, Phylogenetic Analysis, and KA/KS Value

For multiple alignment analysis, the sequences of the conserved domains of each gene were obtained using the Interproscan [78] tool (http://www.ebi.ac.uk/interpro/search/sequence/, accessed on 22 September 2021) and batch CD-search [76] tool in NCBI (https://www.ncbi.nlm.nih.gov/Structure/bwrpsb/bwrpsb.cgi, accessed on 22 September 2021). Using the Geneious V9.1.4 (Biomatters, Auckland, New Zealand) tool, multiple sequence alignments were performed on the sequences of different families. For phylogenetic analysis, the protein sequences of different species were obtained from the NCBI, whereafter neighbor-joining (NJ) was used to construct phylogenetic trees using MEGA-X [79] software with 1000 bootstraps (Appendix A). Evolview [80] (China national center for bioinformation, Beijing, China) was used to enhance the evolutionary tree. To verify the existence of positive selection during the evolution of gene families, the Simple Ka/Ks Calculator tool of TBtools was used to calculate the synonymous substitution rate (Ks) and nonsynonymous substitution rate (Ka) values of homologous gene pairs with their amino acid sequences (Appendix A).

### 3.5. Conserved Motifs, Gene Structure, cis-Element Analysis, and GO Annotation

Conservative motifs were predicted using the online website MEME [81] (http://meme.nbcr.net/meme/Intro.html, accessed on 22 September 2021) and visualized with TBtools v1.089 (Guangzhou, China) [82] software. The exon/intron structures were determined by the pairwise alignment of the full-length cDNA sequences and corresponding genomic sequences on the Gene Structure Display Server 2.0 [83] (http://gsds.cbi.pku.edu.cn, accessed on 22 September 2021). BioEdit software (Los Altos, CA, USA) was used to obtain 1000 bp promoter sequences upstream of corresponding genes in the whole genome data of *S. miltiorrhiza*, and potential *cis*-elements were searched in the PlantCare [84] database. GO annotations were predicted using PANNZER 2 online software [85] (http://ekhidna2.biocenter.helsinki.fi/sanspanz/, accessed on 22 September 2021).

### 3.6. RNA Extraction, cDNA Preparation, and Quantitative Real-Time Reverse Transcription-PCR (qRT-PCR)

The total RNA was extracted from cultured tissues using plant total RNA isolation kits (Aidlab, Beijing, China), and genomic DNA contamination was eliminated during extraction. The integrity of the RNA extraction was verified on 1% agarose gel. The RNA quality and concentration were determined using a NanoDrop 2000C spectrophotometer (Thermo Scientific, Waltham, MA, USA). The total RNA was reverse transcribed into cDNA using Superscript III reverse transcriptase (Invitrogen, Waltham, MA, USA). cDNA was used as a template for gene cloning and qRT-PCR quantification, whereas qRT-PCRs were performed as described previously [86]. *S. miltiorrhiza* β-actin (DQ243702) [87] was employed as an internal reference to detect the expression levels of gene family members in different samples. A real-time quantitative primer was designed using the Real-Time PCR (TaqMan) Primer and Probes Design Tool (https://www.genscript.com/tools/real-time-pcr-taqman-primer-design-tool, accessed on 22 September 2021) website, with the gene-specific primers of qRT-PCR listed in Appendix A. Relative expression analysis data were processed using the 2^−ΔΔCt^ method [88], the variance analysis utilized one-way analysis of variance. For the tissue-specific expression analysis of flowers, leaves, stems, and roots, the transcription level in roots were arbitrarily set to 1, with those in other tissues being set accordingly. For the expression analysis of MeJA and injury treatments, we arbitrarily set the transcription level in the untreated tissues to 1 and compared it with the treated plant tissues. All samples were tested in three technical replicates and three independent biological replicates. The error bar represents the standard deviation of the mean of three biological and three technical replicates.

## 4. Conclusions

For this study, four major genes involved in the JA synthesis pathway in *S. miltiorrhiza* were investigated and analyzed. A total of 23 JA synthesis genes were identified via the systematic analysis of the *S. miltiorrhiza* genome and subsequent molecular cloning. Through the bioinformatics prediction analysis of four enzyme gene families, we found that almost all of them contained MeJA response elements. Furthermore, GO analysis indicates that most of the genes are enriched in the jasmonic acid biosynthetic process of the biological process. The above results are consistent with those of previous studies. However, combined with qRT-PCR results, it was found that some genes had extremely significant changes (more than one hundred times) compared with the control under injury stress, such as *SmAOS2* and *SmOPR3-4*. Under MeJA treatment, *SmLOX9*, *SmAOS3*, and *SmAOS5* showed substantial differences (more than 25 times) compared with the control group. However, the expression levels of some genes in the above two treatments did not change significantly, so we speculated that genes in each family appeared to have functional overlap and redundancy. These results provided many references for elucidating candidate genes for the JA synthesis pathway in *S. miltiorrhiza* and lay a foundation for the future study of gene function. Since these enzyme genes are very responsive to MeJA, we can explore their essential role in the biosynthesis of JA in *S. miltiorrhiza* by developing transgenic lines with overexpressed or mutated enzyme genes. Related studies have shown that increased JA content can promote the synthesis of salvianolic acid, tanshinone, and other medicinal components [21,22,24]. Therefore, this study provides a new strategy for developing excellent strains with high JA content to improve the level of secondary metabolites of *S. miltiorrhiza*.

## Figures and Tables

**Figure 1 ijms-23-09384-f001:**
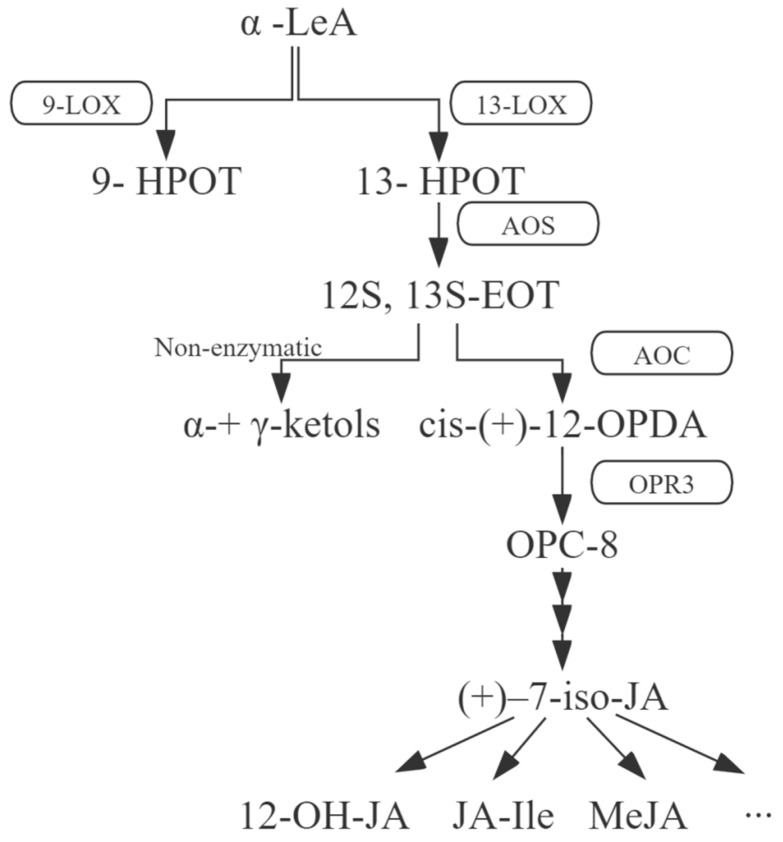
Schematic diagram of the JA biosynthesis pathway. Key enzymes are shown in boxes, with solid black lines and arrows indicating individual biosynthesis steps. The ellipsis indicates a variety of other products. α-LeA, α-linolenic acid; 9-HPOT, 9-hydroperoxylinolenic acid; 13-HPOT, 13-hydroperoxylinolenic acid; 12S, 13S-EOT, 12S, 13S-epoxy-linolenic acid; *cis*-(+)-12-OPDA, *cis*-(+)-12-oxo-phydienoic acid; OPC-8, 8-(3-oxo-2-(pent-2-enyl) cyclopentyl) octanoic acid; α-ketols, 13-hydroxy-12-oxo-octadecadienoic acid; γ-ketols, 9-hydroxy-12-oxo-octadecadienoic acid; 9-LOX, 9-lipoxygenase; 13-LOX, 13-lipoxygenase; AOS, allene oxide synthase; AOC, allene oxide cyclase; OPR3, 12-OPDA reductase3.

**Figure 2 ijms-23-09384-f002:**
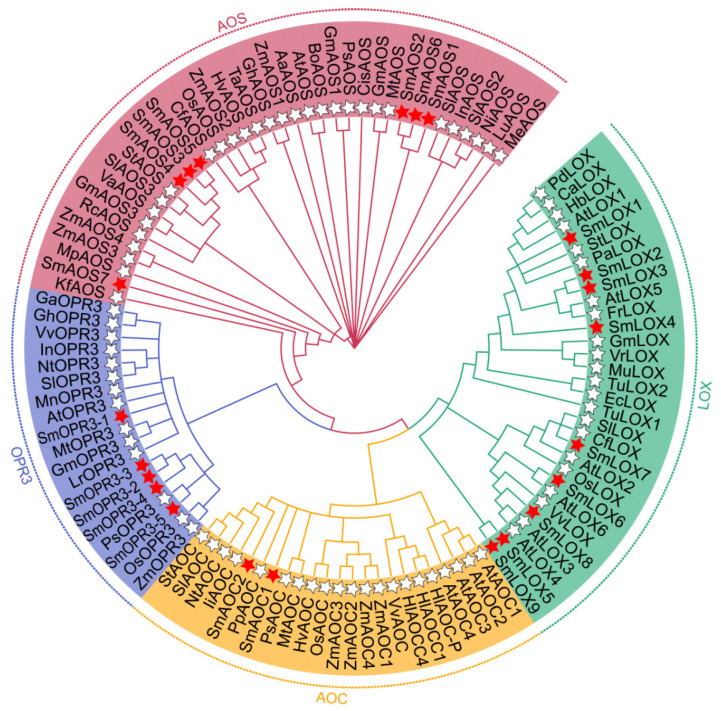
Phylogenetic relationships of JA synthesis pathway enzyme proteins. Amino acid sequences of different plant species were obtained from NCBI based on the accession numbers listed in Appendix A. Red five-pointed stars indicate *S. miltiorrhiza*, and white five-pointed stars indicate other species.

**Figure 3 ijms-23-09384-f003:**
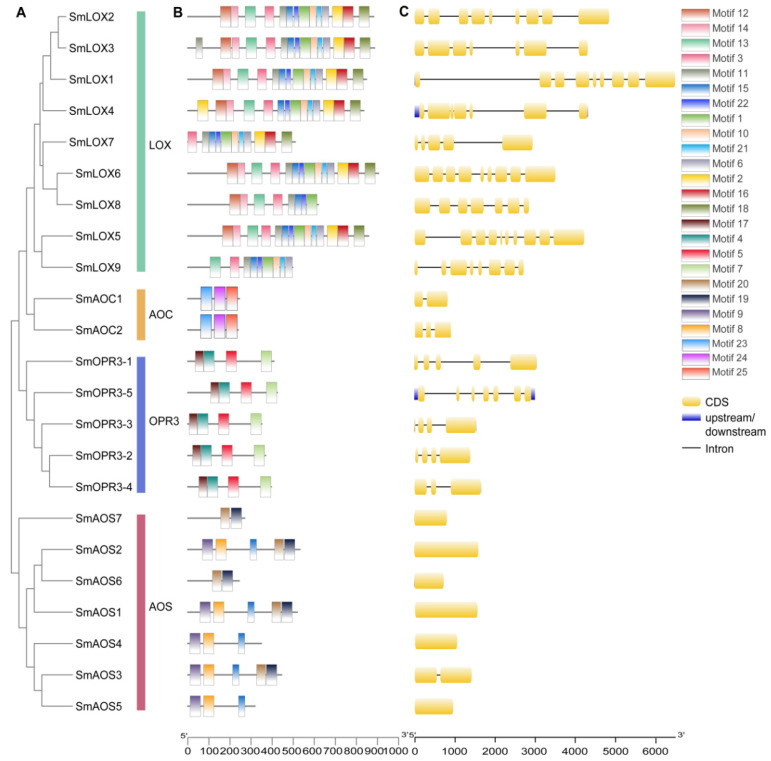
Phylogenetic relationships, gene structures, and conserved motifs of genes related to the JA synthesis pathway in *S. miltiorrhiza*. (**A**) The phylogenetic tree on the left contains 23 enzyme gene proteins. (**B**) Motif patterns of 23 proteins, where each pattern is displayed in a different colored box. (**C**) Exon/intron structures of 23 enzyme genes. Yellow boxes represent exons, black lines represent introns, and blue boxes represent untranslated regions.

**Figure 4 ijms-23-09384-f004:**
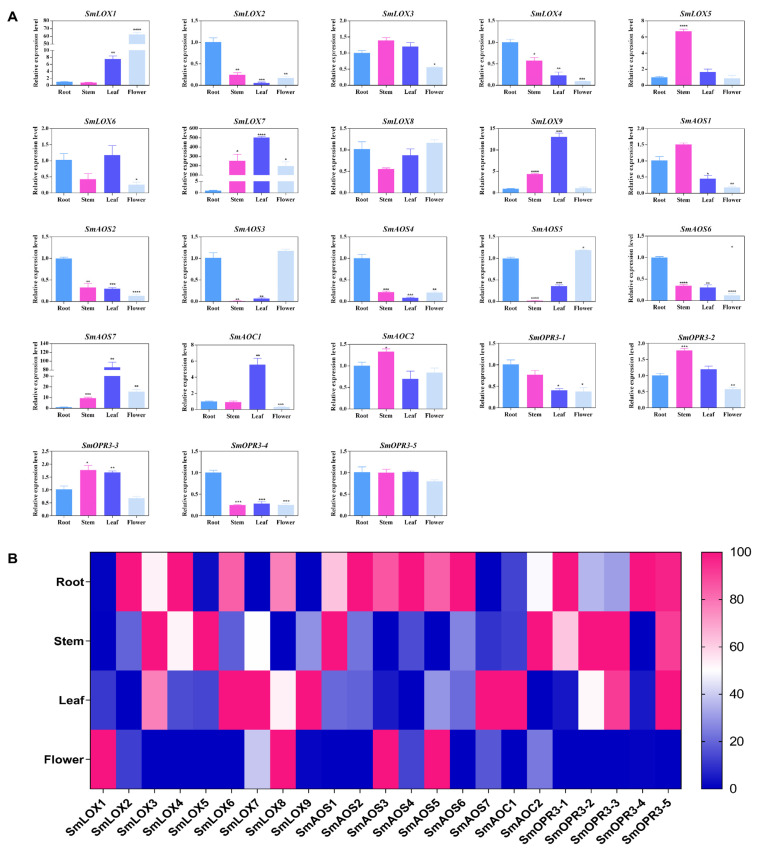
(**A**) Tissue specific expressions of JA biosynthesis related genes. The transcript levels in roots, stems, leaves, and flowers of *S. miltiorrhiza* were analyzed using the quantitative real-time reverse transcription-PCR method (qRT-PCR). Asterisks (*p* < 0.05) indicate significant differences compared with the control group (* *p* < 0.05, ** *p* < 0.01, *** *p* < 0.001, **** *p* < 0.0001). (**B**) Heat map showing expression profiles of 23 enzyme genes in four different tissues. The graph was generated using GraphPad software based on three repeated counts of log2-transformed RNA-Seq data. Red and blue boxes indicate high and low expressions, respectively.

**Figure 5 ijms-23-09384-f005:**
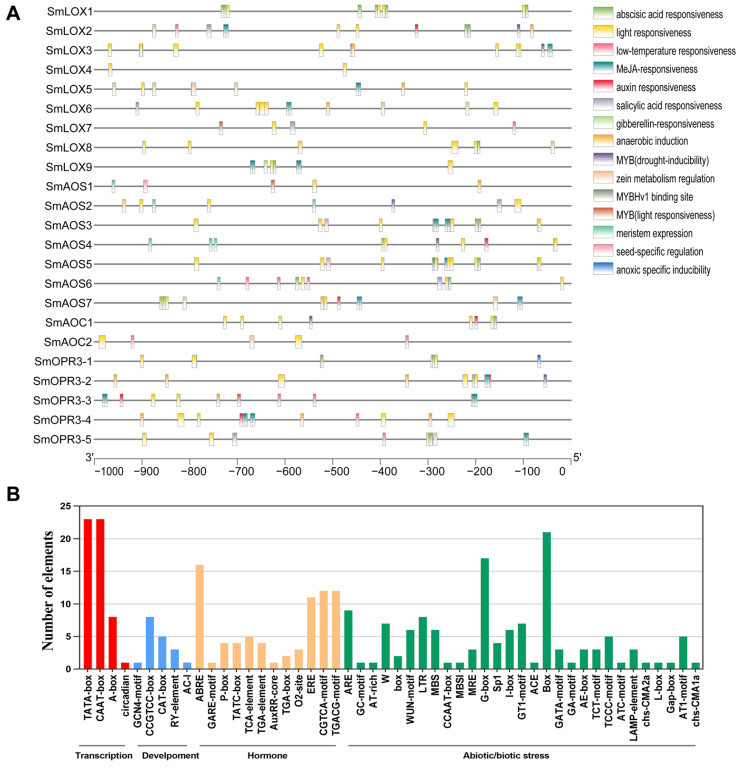
(**A**) Predicted *cis*-elements in the prompters of JA biosynthesis related genes. Distribution of *cis*-elements in the 1 kb upstream promoter regions of JA biosynthesis related genes. Different *cis*-elements are represented by various colors. (**B**) Number of JA biosynthesis related genes containing various *cis*-acting elements. The *cis*-acting elements were identified with the online PlantCARE program using 1 kb upstream of transcription initiation sites of JA biosynthesis related genes. The graph was generated based on the presence of *cis*-acting elements in response to specific elicitors/conditions/processes (*x*-axis) in all JA biosynthesis related genes (*y*-axis).

**Figure 6 ijms-23-09384-f006:**
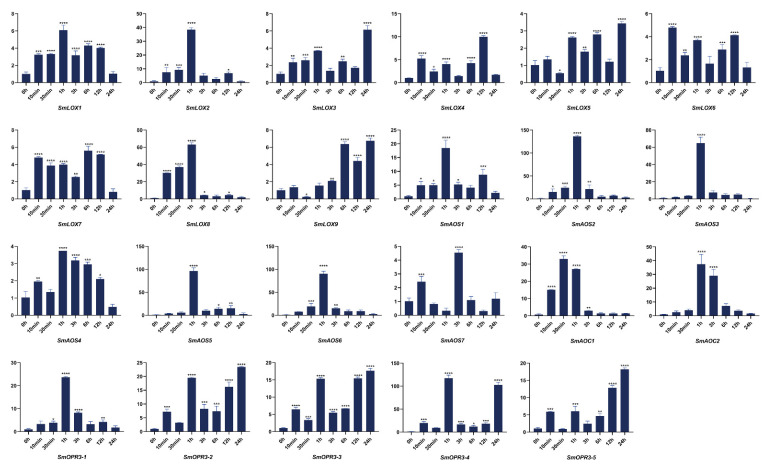
Expressions of JA biosynthesis related genes were correlated at 0 h, 10 min, 30 min, 1 h, 3 h, 6 h, 12 h, and 24 h following the *S. miltiorrhiza* injury treatment. The transcript levels were analyzed using the qRT-PCR method. Asterisks (*p* < 0.05) indicate significant differences compared with the control group (* *p* < 0.05, ** *p* < 0.01, *** *p* < 0.001, **** *p* < 0.0001).

**Figure 7 ijms-23-09384-f007:**
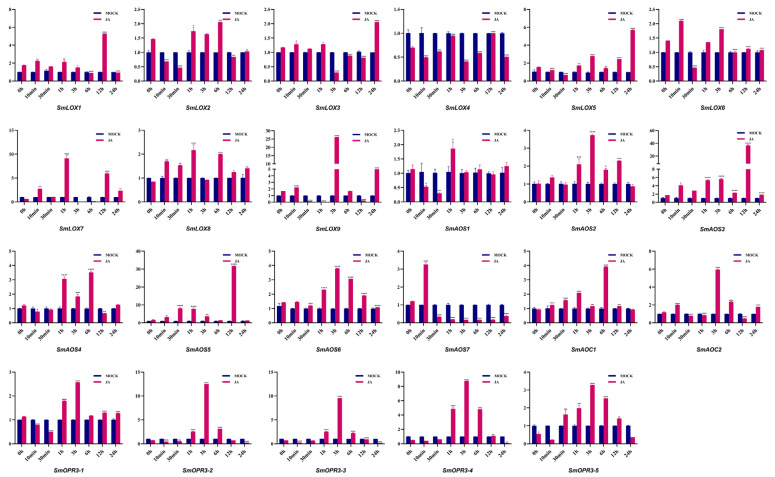
Expression of JA biosynthesis related genes of *S. miltiorrhiza* treated with MeJA for 0 h, 10 min, 30 min, 1 h, 3 h, 6 h, 12 h, and 24 h. The transcript levels were analyzed using the qRT-PCR method. Asterisks (*p* < 0.05) indicate significant differences compared with the control group (* *p* < 0.05, ** *p* < 0.01, *** *p* < 0.001, **** *p* < 0.0001).

**Figure 8 ijms-23-09384-f008:**
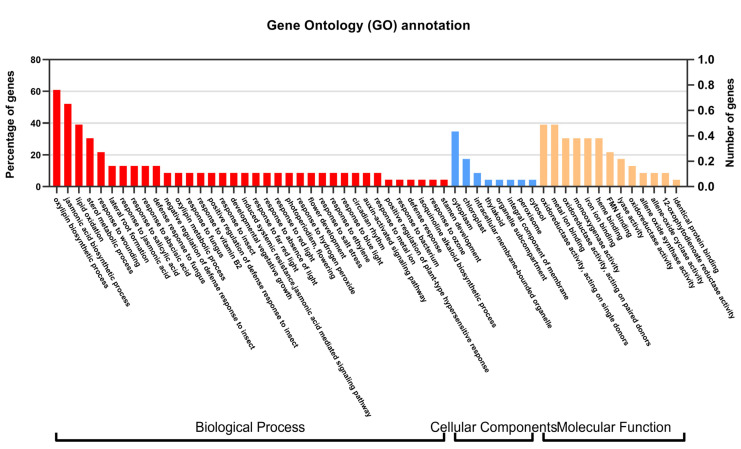
Gene Ontology (GO) annotation of JA biosynthesis related proteins. The *y*-axis on the right side indicates the number of genes in a category and on the left side indicates the percentage of a specific category of genes in the main category. The proteins were divided into three independent categories, namely biological process, cellular components, and molecular function.

## Data Availability

Not applicable.

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
