# Peer review of "Identification and Characterization of Jasmonic Acid Biosynthetic Genes in Salvia miltiorrhiza Bunge"

_ijms, 2022, doi:10.3390/ijms23169384_

Round 1
Reviewer 1 Report
Comment 1: Change the title, instead to "Jasmonic acid biosynthetic enzyme genes" change to "jasmonic acid biosynthesis/biosynthetic genes"
Comment 2: Ln. 15, medicine to medicinal.
Comment 3: Mentioned at line 14, as JA, why again jasmonic and MeJA?
Comment 4: Rewrite the abstract with good English and grammar.
Comment 5: Please upload supplemtary files as word file.
Comment 6: There is a lot of good information, but the conclusion is not up to the level. Rewrite the conclusion.
Comment 7: Fascinating work, but the way of writing not up to the level, please English and Grammar. Also strictly follow the abbreviations throughout the manuscript.
Author Response
Responses to reviewer 1’s comments
Thank you for your letter and for the reviewers’ comments concerning our manuscript entitled “Identification and Characterization of Jasmonic Acid Biosynthetic Enzyme Genes in Salvia miltiorrhiza Bunge” (ID: ijms-1852428). Those comments are all valuable and very helpful for revising and improving our paper, as well as the important guiding significance to our researches. We have studied comments carefully and have made correction which we hope meet with approval. Revised portion are marked in red in the paper. The main corrections in the paper and the responds to the reviewer’s comments are as flowing. We are also ready to further improve the manuscript if any extra comments are made.
Thanks again for your patient help.
All the best,
Zhezhe Wang
Comment 1: Change the title, instead to "Jasmonic acid biosynthetic enzyme genes" change to "jasmonic acid biosynthesis/biosynthetic genes"
Response 1: Thanks for your valuable comments and suggestions. I am sorry for my incorrect description. I have corrected the subject according to your suggestion, and changed it to “jasmonic acid biosynthetic genes” in the revised manuscript (Page1 line 2).
Comment 2: Ln. 15, medicine to medicinal.
Response 2: Thank you for your careful correction. This is my negligence. I have already changed it in the revised manuscript (Page 1 line 16).
Comment 3: Mentioned at line 14, as JA, why again jasmonic and MeJA?
Response 3: Thank you for your accurate comments, and I am sorry for causing trouble to your review. I should have used a more unified reference to the content in the article. In addition, MEJA is actually the short form of methyl jasmonate, which is the downstream product of JA. I'm sorry for the misunderstanding caused by my lack of clear explanation. I have carefully proofread the article before and after, and checked and revised the abbreviations of each paragraph and the whole process.
Comment 4: Rewrite the abstract with good English and grammar.
Response 4: Thanks very much for your professional comments and precious advices. Taking your comments into consideration, I have asked a professional English native speaker to revise my abstract again. I hope you can feel that the revised abstract has reached a better level(Page 1 line 14-25).
Comment 5: Please upload supplemtary files as word file.
Response 5: Thank you for your suggestion. I have changed the attached form into Word format according to your requirements, which was named as “supplementary file”.
Comment 6: There is a lot of good information, but the conclusion is not up to the level. Rewrite the conclusion.
Response 6: Thank you for your full affirmation and objective opinion. After repeated thinking and discussion, we have made a comprehensive summary and outlook of the full text and rewritten the conclusion part (Page 12 line 1576-Page 13 line1611).
Comment 7: Fascinating work, but the way of writing not up to the level, please English and Grammar. Also strictly follow the abbreviations throughout the manuscript.
Response 7: Thank you for your valuable suggestions, and we are sorry for the inconvenience caused to you. We have asked professional native speakers to check and polish the English and grammar of the article again, and at the same time, we have repeatedly checked and unified the abbreviations in the revised manuscript. All changes were marked in red or blue.
Reviewer 2 Report
In this work, four major genes involved in the S. miltiorrhiza pathway of jasmonic acid synthesis were analyzed. A total of 23 genes for the synthesis of jasmonic acid were identified by systematic analysis of the S. miltiorrhiza genome and subsequent molecular cloning. Thanks to bioinformatic predictive analysis of four families of enzyme genes, it was found that the results of S. miltiorrhiza were consistent with previous findings. At the same time, when combined with qRT-PCR results under different conditions, it was found that the genes in each family appeared to have a functional overlap. These results provide a valuable information base with respect to genes for the jasmonic acid synthesis pathway in S. miltiorrhiza. Nevertheless, the authors conclude that the development of transgenic lines with overexpressed or mutated enzyme genes, which have been identified as having significant in vivo correlations, will help to explain many problems with gene functionality. The authors' results of this manuscript provide insight into a new strategy to study the genes for jasmonic acid biosyntase. Therefore, I believe that the work meets the criteria of the journal and can be published.
Author Response
Responses to reviewer 2’s comments
Thank you for your letter and for the reviewers’ comments concerning our manuscript entitled “Identification and Characterization of Jasmonic Acid Biosynthetic Enzyme Genes in Salvia miltiorrhiza Bunge” (ID: ijms-1852428). All of those comments are insightful and helpful for refining and strengthening our paper, as well as providing crucial direction for our research. Response portion are marked in red in the paper. The main corrections in the paper and the responds to the reviewer’s comments are as flowing. If any other comments are provided, we are also prepared to further improve the manuscript.Thanks again for your patient help.
All the best,
Zhezhe Wang
Comment: In this work, four major genes involved in the S. miltiorrhiza pathway of jasmonic acid synthesis were analyzed. A total of 23 genes for the synthesis of jasmonic acid were identified by systematic analysis of the S. miltiorrhiza genome and subsequent molecular cloning. Thanks to bioinformatic predictive analysis of four families of enzyme genes, it was found that the results of S. miltiorrhiza were consistent with previous findings. At the same time, when combined with qRT-PCR results under different conditions, it was found that the genes in each family appeared to have a functional overlap. These results provide a valuable information base with respect to genes for the jasmonic acid synthesis pathway in S. miltiorrhiza. Nevertheless, the authors conclude that the development of transgenic lines with overexpressed or mutated enzyme genes, which have been identified as having significant in vivo correlations, will help to explain many problems with gene functionality. The authors' results of this manuscript provide insight into a new strategy to study the genes for jasmonic acid biosyntase. Therefore, I believe that the work meets the criteria of the journal and can be published.
Response: Thank you very much for your praise and recognition of this article. We feel very honored and grateful for your support and encouragement. We are pleased that you have provided a comprehensive and promising assessment of our work, which has encouraged us and made us more confident about the in-depth scientific research that will follow. Thank you again from the bottom of my heart. I wish you every success in your work and good health.
